# Making Sense of Antisense Oligonucleotide Therapeutics Targeting Bcl-2

**DOI:** 10.3390/pharmaceutics14010097

**Published:** 2022-01-01

**Authors:** Maria Gagliardi, Ana Tari Ashizawa

**Affiliations:** Bio-Path Holdings, Inc., Bellaire, TX 77401, USA; mgagliardi@biopathholdings.com

**Keywords:** antisense oligonucleotides (ASO), apoptosis, Bcl-2, exosomes, liposomes, microRNA (miRNA), small interfering RNA (siRNA), venetoclax

## Abstract

The B-cell lymphoma 2 (Bcl-2) family, comprised of pro- and anti-apoptotic proteins, regulates the delicate balance between programmed cell death and cell survival. The Bcl-2 family is essential in the maintenance of tissue homeostasis, but also a key culprit in tumorigenesis. Anti-apoptotic Bcl-2, the founding member of this family, was discovered due to its dysregulated expression in non-Hodgkin’s lymphoma. Bcl-2 is a central protagonist in a wide range of human cancers, promoting cell survival, angiogenesis and chemotherapy resistance; this has prompted the development of Bcl-2-targeting drugs. Antisense oligonucleotides (ASO) are highly specific nucleic acid polymers used to modulate target gene expression. Over the past 25 years several Bcl-2 ASO have been developed in preclinical studies and explored in clinical trials. This review will describe the history and development of Bcl-2-targeted ASO; from initial attempts, optimizations, clinical trials undertaken and the promising candidates at hand.

## 1. Introduction

The Bcl-2 family of proteins regulates apoptosis, a form of programmed cell death. The family members are highly conserved and have important roles during development and adult tissue homeostasis [1,2,3]. Members of this family, related only by their conserved Bcl-2 homology (BH) domains, are divided into two opposing groups: pro-apoptotic vs. anti-apoptotic [4].

Bcl-2 is the founding member of this family and has anti-apoptotic function. Like other anti-apoptotic members (Bcl-xL, Mcl-1, Bcl-W, Bfl-1, Bcl-b), Bcl-2 has 4 BH domains (BH1 to BH4) (Figure 1A). These domains form a hydrophobic cleft that acts as the binding site for the BH3 domains of their pro-apoptotic counterparts, an interaction which impedes the latter’s function. Pro-apoptotic members are classified as either effectors or initiators of apoptosis. Effectors, Bax and Bak, use their BH domains (BH1 to BH3) to oligomerize and form pores on the outer mitochondrial membrane [5]. Initiators, also known as ‘BH3-only’ proteins, fall into two categories: activators (tBID, Bim, Puma) directly bind to and activate Bax and Bak, whilst sensitizers (Bad, Bik, Bmf, Noxa) bind to anti-apoptotic proteins to disrupt their sequestration of BH3 activators and interactions with Bax and Bak [4,5] (Figure 1A).

The intrinsic apoptotic pathway involves the permeabilization of the outer mitochondrial membrane, and release of cytochrome C and caspase activating proteins via the pores formed by Bax and Bak. Apoptotic stimuli, such as oxidative stress, calcium flux, ER stress and DNA damage, disrupt the steady-state interactions between pro- and anti-apoptotic proteins that are normally kept in fine equilibrium [6]. It is the relative changes in protein abundance and affinities that ultimately determine cell fate [7] (Figure 1B).

Cancer cells advantageously manipulate the apoptotic machinery to resist cell death [8]. Bcl-2 itself was discovered due to its overexpression in B cell lymphomas as a result of chromosomal translocation t(14;18), in which the *bcl-2* gene on chromosome 18 fuses to the immunoglobulin heavy-chain (IgH) gene on chromosome 14 and is expressed under the control of the IgH enhancer [9,10,11]. Over the past 25 years in addition to hematological malignancies, elevated Bcl-2 expression and function has been observed in various tumors, including melanoma [12], breast cancer [13,14], colorectal cancer [15] and lung cancer [16]. Mechanisms of overexpression include gene amplification [17], somatic mutations at the *bcl-2* promoter [18], transcriptional upregulation [19] and loss of microRNA (miRNA) that negatively regulate Bcl-2 post-transcriptionally [20].

Beyond evading cell death, the BH4 domain of Bcl-2 gives it the ability to interact with non-apoptotic proteins and take part in other cellular processes [21]. Angiogenesis is an important part of tumor growth and metastasis. Solid tumors rely on the formation of blood vessels for continued supply of nutrients, oxygen and as a route in the metastatic pathway [22]. Vascular endothelial growth factor (VEGF) is a key mediator of angiogenesis that promotes the differentiation, proliferation and migration of endothelial cells [23,24,25]. VEGF and Bcl-2 have a synergistic relationship [24,26]. Both in vitro and in vivo studies have shown that elevated Bcl-2 level in cancer cells under hypoxia condition induces VEGF expression, enhances its mRNA stability, promotes its secretion and its transcriptional activity, resulting in increased vascularization that is independent of cell survival [24,27]. Overexpression of Bcl-2 lacking the BH4 domain has no effect on VEGF expression/activity, making the BH4 domain the key region for Bcl-2’s crosstalk with angiogenesis [27,28]. The BH4 domain has also been shown to play a part in the upregulation of NF-κB transcriptional activity through Raf-1/MEKK1-mediated activation of IKKβ [14,29,30]. NF-κB is an important regulator of epithelial-mesenchymal genes (*N-cadherin*, *vimentin*) and matrix metalloproteinase (*mmp-9*, *mmp-2*), all required for cancer metastasis [29,31,32]. Similarly, under hypoxia conditions Bcl-2 acts as a cofactor that facilitates nuclear transport of transcription factor twist1; once again encouraging epithelial-mesenchymal transition, angiogenesis and tumor development [25].

Given its multifunctional and dynamic roles in tumorigenesis, it is not surprising that Bcl-2 has been a popular target in cancer drug design. Although Bcl-2 has crucial functions in non-cancerous cells, therapies targeting Bcl-2 have a predominant effect on tumor cells. This could be due to the fact that tumor cells are ‘primed’ for apoptosis as a result of DNA damage, hyper proliferation and aberrant oncogenic signaling [33,34]. Normal cells are not primed for apoptosis so a decrease in Bcl-2 expression does not necessarily result in cell death. Indeed, many adult tissues, such as the heart, brain, kidneys and liver are refractory to apoptosis [35]. Hematopoietic cells are not refractory to apoptosis and express varying levels of Bcl-2 depending on their population, developmental stage and activation [36]. Yet, studies had shown that inhibition of Bcl-2 using ABT-737, a BH3 mimetic, significantly reduced the viability of multiple myeloma cells with no effect on normal peripheral blood mononuclear cells [37]. It is possible that the ratio between pro- and anti-apoptotic proteins and the specific dependency on Bcl-2 for cancer cell survival permits the establishment of a therapeutic index and safe targeting of Bcl-2.

Oligonucleotide therapeutics, specifically antisense oligonucleotides (ASO) therapeutics, are an eminent drug platform with advantages in terms of specificity, production, and delivery [38]. These short (18–25 bases in length), chemically modified nucleic acid polymers rely on Watson-Crick base pair binding to target mRNA, resulting in either gene silencing, steric hindrance, or alternative splicing [39]. There are currently six FDA approved ASO drugs [40]. These approved drugs fall into three generations of modified ASO, which display the optimization strategies used. First generation ASOs have a phosphorothioate (PS) rather than phosphodiester (PO) backbone, to increase cellular uptake and nuclease resistance [41]. Second generation ASOs have PS modified central regions with methyl or methoxyethyl modifications at the 2′ position of the sugar moieties at the 3′ and 5′ ends (2′-O-methyl (2′-OMe) or 2′-O-methoxyethyl (2′-MOE)). These gapmer structures further increase hybridization and nuclease resistance [42,43,44]. Similar to second generation, third generation ASOs also have a gapmer configuration but with more complex modification at the 3′ and 5′ ends. These include: (i) locked nucleic acids (LNA) in which a methylene bridge connects the 2′ oxygen and 4′ carbon of the riboses; (ii) phosphorodiamidate morpholino oligomers (PMOs) in which the ribose is replaced by morpholine moiety and nucleotides are connected via methylene phosphoramidate bonds; (iii) peptide nucleic acids (PNAs) in which the deoxyribose phosphate backbone is replaced by a pseudopeptide polymer [45,46].

In this review the history of ASO targeting Bcl-2 will be described. By recognizing their challenges and appreciating their optimization it becomes clear why ASO remain a promising Bcl-2 therapeutic.

## 2. The Beginning of Bcl-2 ASO Therapies

Identified as an oncoprotein in the mid-1980s [47] and as a mediator of chemotherapy resistance in the mid-1990s [48,49,50], Bcl-2 has high potential as a cancer therapeutic target. Since Bcl-2 is an intracellular protein with no enzymatic activity, Bcl-2 was not amenable to the conventional therapeutic approaches at that time, which consisted of antibodies targeting cell surface proteins and small molecules inhibiting enzyme activities [51]. This led to utilizing ASO technology to inhibit Bcl-2 expression. The evolution of ASO technology has been accompanied by increased efficacy in Bcl-2 regulation with changes in target sequence selection, polymer backbone modifications and delivery strategies.

### 2.1. Initial Bcl-2 ASOs In Vitro Studies

Initial studies exploring the ability of Bcl-2 ASO to inhibit leukemia cell survival used a 20-base pair oligonucleotide complementary to the transcription initiation site of Bcl-2 mRNA [52]. A comparison between oligonucleotides with PO vs. PS backbones, in the absence of delivery technology, revealed the latter to be more resistant to nucleases and 5–10 times more potent at inhibiting 697 cell (human pre-B-cell acute lymphocytic leukemia) proliferation [52]. Although both oligonucleotides initially suppressed cell proliferation via non-cytotoxic mechanisms, with inhibition of cell replication preceding cell death, their rate of cellular uptake and kinetics differed. In serum-free conditions, PO-ASO achieved maximum intracellular concentrations within two days after initial treatments and maximum cell inhibition within two to three days, whereas PS-ASO required four to five days to achieve comparable intracellular levels and four to seven days to achieve maximum inhibition [52]. Similar results were obtained in human leukemic myeloid cell lines: Daudi, KG1a and HL-60 [53]. Treatment of acute myeloid leukemia (AML) patient cells in the presence of serum with PS-ASO showed increased sensitivity to cytostatic drugs: daunorubicin or l-P-D-arabinofura-nosyl-cytosine (ara-C) [53], thereby establishing Bcl-2 PS-ASO as more clinically relevant.

The dependency of ASO’s effectiveness on which Bcl-2 region is targeted was investigated using 13 different ASO directed against Bcl-2 coding regions, 5′ or 3′ untranslated regions and translation initiation or termination sites. Amongst these, the ASO targeting codon 141–147 of Bcl-2 mRNA (ASO-2009) proved most cytotoxic in a human small cell lung cancer (SCLC) cell line, SW2 [54]. Contrary to the PS-ASO described above, ASO 2009’s inhibitory activity was apparent after only two days. This could be due to its facilitated intracellular delivery with cationic lipids [54]. Combination treatments of ASO-2009 with etoposide, doxorubicin or cisplatin in SCLC cell lines displayed synergistic cytotoxicity, making them a novel treatment strategy [55].

### 2.2. Bcl-2/Bcl-xL-Bispecific ASO

In addition to Bcl-2, upregulation of anti-apoptotic Bcl-xL is also common in various cancers and is associated with resistance to therapeutics [56,57,58,59]. While being functionally redundant and structurally similar to Bcl-2, Bcl-xL may protect cells from different apoptotic stimuli [60]. Cancer cells can take advantage of this and use Bcl-xL as an alternative to, or in conjunction with, Bcl-2 to survive [61,62]. Upon identifying a region of high sequence homology in the Bcl-2 and Bcl-xL mRNAs (nucleotides 605–624 and 687–706, respectively, which differs by only three nucleotides), three bi-specific ASOs targeting Bcl-2 and Bcl-xL were designed to simultaneously inhibit both survival factors. These 20-base pair ASO are classified as second generation with 2′-MOE modification at their ends and hold a maximum of three mismatching bases [63]. Analysis in SCLC and non-small cell lung cancer (NSCLC) cell lines showed that ASO-4625, which has no mismatching nucleotides to Bcl-2 but three mismatching nucleotides to Bcl-xL, had the strongest bispecific activity, inhibiting the expression of both proteins and inducing cell death [63]. Such activity was also confirmed in breast, colorectal, prostate and malignant melanoma cells [58,59]. In vivo studies proved ASO-4625’s therapeutic potential, with a 51% reduction of breast cancer (MDA-MB-231) xenografts and 59% reduction in colorectal cancer (Colo-320) xenografts after a 3-week treatment period [64]. In addition to restoring apoptosis, treatment of melanoma cells with ASO-4625 led to a 50–60% reduction in VEGF secretion and a 10–12 fold decrease in the vascularization of matrigel plugs subcutaneously injected into mice; reiterating Bcl-2’s relevance in angiogenesis and implicating Bcl-xL’s role in it [65]. ASO-4625 also markedly increased the sensitivity of prostate cancer cells to mitoxantrone, docetaxel and paclitaxel, suggesting its use in combination treatments [66,67].

To optimize the binding of ASO-4625 to Bcl-xL and overcome the potential destabilization due to mismatches, an isosequential analog with LNA, ASO-5005, was developed, [68]. ASO-5005 simultaneously downregulated Bcl-2 and Bcl-xL expression and reduced cell viability more effectively than ASO-4625 [68]. Combination treatments of ASO-5005 with doxorubicin or paclitaxel in MDA-MB-231 cells and cisplatin, paclitaxel, or gemcitabine in H125 cells (NSCLC) were stronger than those of chemotherapies alone or in combination with ASO-4625 [68]. Unfortunately, the investigators for ASO-4625 and ASO-5005 were not able to identify suitable pharmaceutical/business partners to help develop these drug candidates. Despite these encouraging results the above ASO did not progress beyond in vitro and in vivo preclinical studies.

### 2.3. Oblimersen

Oblimersen (also known as: G3139, Genasense, Oblimersen sodium; from Genta Inc., Berkeley Heights, NJ, USA) was the first Bcl-2 ASO to enter clinical trials [69,70]. It obtained orphan drug status in both the US and EU in 2001 for chronic lymphocytic leukemia (CLL) and multiple myeloma [71]. This 18-base pair PS-ASO (5′-TCT CCC AGC GTG CGC CAT-3′) binds to the first six codons of the Bcl-2 mRNA [72]. Preclinical evidence indicated that Oblimersen enhanced the in vivo efficacy of chemotherapeutic agents against hematological malignancies and solid tumors. In a lymphoma xenograft model, Klasa et al. showed that the median survival of SCID mice inoculated with 5 million DoHH2 lymphoma cells was 44 days, with 0% of mice surviving beyond 90 days [73]. In this model, the median survival of mice administered with 5 mg/kg of Oblimersen or 35 mg/kg cyclophosphamide were 79 days and 47 days, respectively; percentages of mice administered with Oblimersen or cyclophosphamide that survived >90 days were 48% and 0%, respectively. The median survival of mice treated with the Oblimersen and cyclophosphamide combination was not obtained as 61% of mice survived >90 days [73]. Similarly, Oblimersen (5 mg/kg) was reported to enhance docetaxel (20 mg/kg) efficacy in a preclinical H-460 lung cancer model [74]. Tumor growth delay by Oblimersen, docetaxel, and combination were 5 days, 13 days, and 21 days, respectively. Four of seven mice achieved complete responses with the combination treatment, whereas no complete response was observed in mice treated with Oblimersen or docetaxel alone [74].

Following its successful preclinical antitumor activity, Oblimersen has since been in 45 clinical trials (https://clinicaltrials.gov/ last accessed 30 October 2021). Since minimal clinical activity was observed in Oblimersen single agent phase I/II studies [75,76,77,78], subsequent phase III clinical trials for patients with hematological malignancies or melanoma, involve Oblimersen in combination with other chemotherapeutic drugs such as dacarbazine [79], dexamethasone [80], cyclophosphamide [80] and fludarabine [81]. An earlier phase I study reported that addition of Oblimersen to standard cytarabine/daunorubicin chemotherapy led to complete remission (CR) in 14 of 29 untreated AML patients over the age of 60 [82]. Among 13 patients assessable for both clinical response and Bcl-2 protein levels, approximately a 20% decrease in Bcl-2 protein levels was observed in patients with CR (*n* = 7) but no changes in Bcl-2 protein level in non-responders (*n* = 6). Although the number of samples was small, patients’ response appeared to correlate with baseline Bcl-2 levels; among 22 patients assessable for both clinical response and Bcl-2 mRNA levels, median number of Bcl-2 mRNA copies were higher in the CR patients (*n* = 12) than those in the non-responders (*n* = 10). These results led to the phase III Cancer and Leukemia Group B (CALGB) 10201 study in which the effect of adding Oblimersen to standard cytarabine/daunorubicin chemotherapy in previously untreated AML patients over the age of 60 was assessed and evaluated based on the rates of CR, disease free survival (DFS) and overall survival (OS). Of the 506 patients enrolled those that received combination treatment displayed no statistically significant improvement in CR, DFS or OS [83]. A subset analysis of patients with secondary AML, however, did display improved DFS with the addition of Oblimersen to their treatments [83]. It is not clear why Oblimersen combination treatment failed in the CALGB 10201 study. This might be due to the fact that baseline Bcl-2 levels were not used to stratify enrolled patients, target Bcl-2 inhibition was not sufficient, and/or other disease mechanisms that precluded Oblimersen from inducing a response. Retrospective subset analysis of a randomized phase III study showed that combining Oblimersen with dacarbazine significantly improved OS and progression-free survival (PFS) in chemotherapy-naïve patients with advanced melanoma and normal baseline serum lactate dehydrogenase (LDH) levels [79]. But a prospective, double-blind, placebo-controlled study did not confirm that addition of Oblimersen to dacarbazine could significantly improve OS or PFS in chemotherapy-naïve patients with advanced melanoma and low-normal LDH levels [84]. The authors noted that advanced melanoma patients with low-normal LDH levels might represent a population with a better chance of benefitting from systemic therapy, thus obscuring the earlier analysis and revealing the weakness of retrospective subgroup analysis of clinical studies. The inability of these combination treatments to affect OS, which is the primary outcome measure in these studies, puts the efficacy of Oblimersen in question and denies its FDA approval [85] (Figure 2).

The overall failure of Oblimersen in phase III trials may be due to instability, limited cellular uptake or intracellular compartmentalization. This has encouraged the optimization of Bcl-2 ASO structure and delivery.

## 3. Optimizations of Bcl-2 ASOs

### 3.1. SPC2996

SPC2996, developed by Santaris Pharma, is a 16 PS-base pair oligonucleotide with two LNA moieties at each end [86] (Figure 3A). It differs in sequence from Oblimersen by only 3 nucleotides [86]. Delivered to cells gynomically (i.e., without conjugates or transfectants), it was efficiently internalized by a range of cell lines (melanoma, lymphoma and fibrosarcoma) and was more potent than Oblimersen at reducing Bcl-2 expression [86]. CLL patients in a SPC2996 phase I/II clinical trial did not achieve significant clinical remission despite having a dose-dependent reduction in circulating CLL cells [87]. Treatment caused dose-dependent inflammatory reactions in 39% of patients, a side effect also seen in Oblimersen trials [87]. Transcriptomic profiling of peripheral blood samples revealed upregulation of 466 genes, many of which are involved in the activation and regulation of immune responses. Toll like receptor (TLR) early response genes such as IL-1β, MIP-1α and TNF-α were particularly overexpressed. These findings support the hypothesis that TLR9 is activated by the CpG motifs in the SPC2996 and Oblimersen sequences, explaining their paralleled and concomitant inflammatory effects [88,89].

### 3.2. PNT2258

Developed by ProNai therapeutics, now known as Serra Oncology, PNT2258 optimized the silencing of Bcl-2 in two ways: (i) the target sequence selection, and (ii) the delivery mechanism. Rather than hybridizing to Bcl-2 mRNA, PNT2258 adopted the DNA interference (DNAi) approach in which a non-coding regulatory region upstream of the transcription start site is targeted [90]. The regulation of gene expression by distal elements is complex and includes the activity of enhancer, modulators of chromatin structure, insulators, locus control regions and promoters [91,92]. Indeed, in t(14;18) lymphoma cells, aberrant expression of *bcl-2* is a result of IgH 3′ enhancers interacting with *bcl-2*’s promoter regions [93]. The 24-base phosphodiester oligonucleotide used in PNT2258, PNT100, binds to a sequence in *bcl-2*’s P1 promotor region which is recognized by transcription factor Sp1 [94]. Obstructing the interaction of the promotor with the transcriptional machinery abrogates *bcl-2* expression. This unmodified oligonucleotide is delivered to cells in an amphoteric liposome (SMARTIClES^®^, [95]) composed of 1-Palmitoyl-2-oleoyl-sn-glycero-3-phosphocholine, 1,2-dioleoyl-sn-glycero-3-phosphoethanolamine, cholesterol hemisuccinate, and cholesteryl-4-([2-(4-morpholinyl) ethyl]amino)-4-oxoburanoate [96] (Figure 3B). These pH-sensitive liposomes are anionic at neutral pH and cationic under acidic conditions. Their anionic state averts the aggregation of serum components thereby promoting serum stability and reducing toxicities. Their cationic state mediates efficient encapsulation of oligonucleotides during production and fusion with endosomes, thereby enabling endosomal escape within the cells [97].

Preclinical studies in non-Hodgkin’s lymphoma (NHL) cell lines (WSU-FSCCL, WSU-DLCL2 and WSU-WM) showed that PNT2258 treatments decreased cell viability in a dose-dependent manner [94]. This effect was more significant in cell lines with the t(14;18) translocation (WSU-FSCCL, WSU-DLCL2), supporting the notion that since high Bcl-2 levels in these cells was dependent on increased transcription and its distal regulation, PNT100’s activity had a greater impact [94]. Apoptotic markers such as cell shrinkage and nuclear chromatin condensation were seen 72 h after treatments with concentrations as low as 2.5 µM. DNA fragmentation, evident in the late stages of apoptosis, increased from 20% after 48 h to 40% after 72 h [94]. In addition to apoptosis, PNT2259 also induced cell cycle arrest in a non-Bcl-2, rather CDK4-dependent mechanism [98]. The exposure of cells to 2.5 µM PNT22598 increased the proportion of cells in the sub-G0 and G0/G1 phases and decreased those in G2/M [94]. The efficacy of the liposomal carrier was evidenced by the stability of PNT2285 in whole blood for at least 24 h and the absence of PNT100 in plasma post intravenous administration of PNT2258 in mice [96]. The antitumor activity of PNT2258 was tested in WSU-DLCL2, Daudi-Burkitt’s (lymphoma), PC3 (prostate) and A375 (melanoma) tumor-bearing mice. In the WSU-DLCL2 model PNT2258 demonstrated effective activity as a single agent and even more so in combination with Rituximab. Rituximab combination treatments also had a dramatic impact on tumor growth in Daudi xenografts as did Docetaxel combination treatments in A375 xenografts; showing PNT2258′s potential as an addition to standard treatments in non t(14;18) and highly resistant tumors [96].

A phase I clinical trial in patients with advanced solid tumors (NCT01191775) confirmed the high tolerability and low toxicity of PNT2258 at doses up to 150 mg/m^2^ [99]. Delivery was systemic and 27% of the patients receiving PNT2258 benefited with disease stabilization [99]. In a phase II clinical trial (NCT01733238) 13 patients with relapsed/refractory B-cell malignancies received PNT2258 monotherapy and had overall response rate of 53.8% and duration of response of 23.4 months [100]. Unfortunately, such results were not replicated in a larger phase II trial (Wolverine, NCT02226965). After a response rate of only 8.1% in patients with relapsed/refractory diffuse large B cell lymphoma, ProNai halted PNT2258’s development [101].

### 3.3. BP1002

BP1002 is a novel liposomal 20-base pair ASO which targets Bcl-2 mRNA at the translation start site. Its P-ethoxy backbone, a hydrophobic analog of phosphodiesters, makes it nuclease-resistant and lipophilic. The liposomes are composed of dioleoylphosphatidylcholine (DOPC) phospholipids and Tween 20 surfactant. DOPC is non-immunogenic and with BP1002’s overall neutral charge, potential toxicities due to interactions with cellular proteins of plasma proteins are reduced. Pharmacokinetic and tissue distribution studies of L-Bcl-2 (pilot name for BP1002) in mice caused no adverse effects and had primary accumulation in the liver and spleen [102]. As these are the primary organs in which leukemias and lymphomas manifest, preclinical studies of L-Bcl-2 have focused on such disease models.

Bcl-2 expression in HL-60 (AML) cells decreased significantly with L-Bcl-2 treatments, as evidenced by western blot and flow cytometry. The latter showed a 39% decrease in Bcl-2 expression after three days of treatment and a 60% decrease by day five [103]. Cell viability also decreased significantly and was mirrored by an increase in apoptotic sub G1 cells [103]. Similarly, L-Bcl-2 decreased proliferation of lymphoma cells lines [104]. Combination treatments of L-Bcl-2 with ara-C also proficiently induced cytotoxicity in doxorubicin-resistant HL-60 cells, which express elevated levels of both Bcl-2 and Bcl-xL [105,106]. In primary AML patient samples, 57.9% showed a decrease in viability and increase in apoptosis [105]. However, only in patient samples with low levels of Bcl-2 was L-Bcl-2 able to increase the sensitivity to ara-C [105]. Given that low Bcl-2 levels had been associated with shorter remission in AML patients with unfavorable cytogenetics, it is encouraging to think that L-Bcl-2 treatments may serve to increase their sensitivity to different chemotherapies [107]. BP1002 toxicology studies involved the treatment of mice with 7.5, 15 or 30 mg BP1002/kg and the treatment of rabbits with 3.75, 7.5, or 10 mg BP1002/kg biweekly for four weeks. In both animal models no changes in animal weight, hematological parameters (platelet count and coagulation profile) or hepatic and renal functions were observed. Thrombocytopenia is a commonly reported dose-limiting toxicity in patients administered with ASOs in clinical trials and has led to drug withdrawals [108,109]. Although the exact mechanism by which ASOs cause a reduction in platelets is still under investigation [108], the high tolerability of BP1002 in preclinical studies is promising and may be due to its P-ethoxy structure [110]. Mice harboring CJ cell xenografts (B-cell lymphoma cells) that were injected with 20 mg/kg of L-Bcl-2 biweekly for five weeks displayed delayed disease progression and had an 80% survival rate. Untreated or empty liposomes treated mice had 20% and 0% survival, respectively, by week five [104]. Establishing BP1002’s preclinical active dose to be 10 to 20 mg/kg in mice, it is estimated to be between 30 to 60 mg/m^2^ in humans [111].

A phase I clinical trial is currently enrolling patients with advanced lymphoid malignancies to evaluate the safety, pharmacokinetics and efficacy of BP1002 monotherapy (NCT0407258). Escalating doses of BP1002, starting at 20 mg/m^2^, will be intravenously administered biweekly for four weeks. BP1002 is currently the only Bcl-2 specific ASO in clinical trial (Figure 2).

## 4. Outlook on Bcl-2 ASO and Alternative Oligonucleotide Therapies

### 4.1. Overcoming Venetoclax Resistance

After years of intensive research, small molecules that can target the Bcl-2 family were developed. Venetoclax (ABT199) is the only FDA approved Bcl-2 inhibitor. This BH3 mimetic obstructs Bcl-2’s BH3 binding groove, making Bcl-2 unable to sequester and neutralize pro-apoptotic proteins [112]. Although Venetoclax has provided a much-needed step forward in cancer treatments, with a successful impact on patients with CLL, small lymphocytic leukemia (SLL) and AML, limitations such as toxicities and drug resistance remain prominent [113,114]. A detailed description of drug resistance mechanisms are provided in several recent reviews [115,116,117,118]; they include: (i) the compensatory upregulation of anti-apoptotic proteins Bcl-xL and Mcl-1 [119,120,121], (ii) super enhancer associated transcriptional reprogramming [116], (iii) enhanced mitochondrial architecture and translation [122,123] and (iv) Bcl-2 point mutations that inhibit Venetoclax binding (G101V, D103Y) [115,124]. A common approach to improve or restore the efficacy of Venetoclax is combination treatments, summarized by Yue et.al [117]. These treatments involve more than 2 different drugs and carry elevated risks of hematological and gastrointestinal toxicities [117]. Recently, the ability of BP1002 to overcome venetoclax resistance in leukemia and lymphoma cells (MV-4-11, SU-DHL-2, SU-DHL-6) in combination with decitabine has been assessed. Preliminary in vitro results show that the BP1002 + decitabine combination is more effective at decreasing cell viability in Venetoclax-resistant cells than the Venetoclax + decitabine combination [125]. Although BP1002 combination treatment studies in vivo and a phase I clinical trial are currently underway, it is encouraging to know that in a phase 1 clinical trial of BP1001 (a sister drug targeting Grb2, differing only by the ASO sequence) patients did not experience any significant dose limiting toxicities, making the maximum tolerated dose yet undetermined [126]. BP1002 may therefore be a more tolerable and efficient alternative to Venetoclax in combination treatments for Venetoclax-resistant malignancies. Nonetheless, the infancy of BP1002 makes it difficult to directly compare it to Venetoclax. While BP1002’s low toxicity is certainly advantageous and holds great promise as an adjuvant therapy, BP1002’s ability to overcome Venetoclax resistance will depend on the resistance mechanism(s) employed. For example, ASO mediated inhibition of Bcl-2 might be of little use in cells that are refractory to Venetoclax due to elevated Mcl-1 expression or reduced Bax expression. On the other hand, BP1002 might overcome Venetoclax resistance if the mechanism is due to acquired mutations in Bcl-2’s BH3 binding groove. This is due to the fact that BP1002 and Venetoclax target Bcl-2 differently; BP1002 targets the translational initiation region of the Bcl-2 mRNA while Venetoclax targets the BH3 domain of the Bcl-2 protein. Recently, Venetoclax has been shown to have off-target metabolic effects that are independent of its Bcl-2 targeting activity. Venetoclax treatment can alter mitochondrial morphology and reduce mitochondrial respiration even in cells with CRISPR/Cas9 mediated Bcl-2 deletion [127]. These changes are dependent on the transcription factor ATF4 and may promote a cell’s sensitivity to Venetoclax [127]. Off-target effects of BP1002 have not yet been observed and will be investigated.

### 4.2. Alternative Oligonucleotide Therapies

The regulation of Bcl-2 holds such therapeutic potential that in addition to ASOs, RNA based therapies are being developed to target it.

#### 4.2.1. MRX34

MiRNA are endogenous 20–25 nucleotide non-coding RNAs that act as mRNA regulators. When bound to the 3′UTR of their target mRNAs, miRNA suppress target translation and promote target mRNA degradation [128,129]. Discovered in the early 1990’s in *C. elegans* [130], they were quickly identified across species and are now known to regulate >60% of human protein coding genes [131]. With such an important and delicate role in gene expression it is not surprising that miRNA dysregulation is seen in tumorigenesis and drug resistance [132,133,134]. MiRNA-34a is a tumor suppressor miRNA of the miRNA-34 family. Regulated by transcription factor p53, which mediates anti-proliferative stress responses, miRNA-34a suppresses the expression of more than 200 targets that have regulatory roles in cell cycle, metastasis, epithelial-mesenchymal transition, stemness, invasion and cell survival [135,136,137]. Bcl-2 is a miRNA-34a target, thus the reduced expression of miRNA-34a in various solid and hematological malignancies is often accompanied by an increase in Bcl-2 expression [138,139,140,141]. Mirna Therapeutics, Inc generated a miRNA-34a mimetic which was incorporated into a liposome; the drug product was named MRX34 [142]. Preclinical in vivo studies reported systemic delivery, low toxicity and sufficient suppression of tumor growth in mouse models with lymphoma, liver and lung cancer [142,143,144]. A phase I clinical trial (NCT02862145) however, had to be withdrawn and MRX34 development halted after severe immune mediated adverse effects resulted in the death of four patients [145].

#### 4.2.2. TargomiR

MiRNA-15/16 is another family of tumor suppressing miRNA that regulates Bcl-2 levels and has reduced expression in various cancer [146,147]. Malignant pleural mesothelioma (MPM) patients samples revealed a 22-fold decrease of miRNA-16 in tumor tissues compared to normal tissues, a 4-fold decrease of miRNA-15a, and a 10-fold decrease of miRNA-15b [148]. Transfection of MPM cells with a miRNA-16 mimic most effectively reduced cell growth and colony formation, led to the accumulation of cells in G0/G1 and increased cell sensitivity to chemotherapeutic drugs [148]. Treatment of tumor xenografts with mimic miRNA-16 resulted in dose-dependent inhibition of tumor growth. MiRNA-16 is delivered to the tumor cells in a nano vehicle named TargomiRs. These are nonliving bacterial minicells, which have intrinsic endosomal escape mechanisms, coated with EGFR antibodies to enable targeting [148] (Figure 3C). In a dose escalation phase I clinical trial (NCT02369198) TargomiR was well tolerated in patients with MPM and NSCLC, with no adverse immune effects and early signs of antitumor activity [149]. These results encourage further trials in combination with chemotherapy or immune checkpoint inhibitors (Figure 2).

#### 4.2.3. siBCL-2 NKExos

As shown in the examples above the development of oligonucleotide therapeutics is accompanied by innovative delivery technologies: from neutral liposomes to pH sensitive liposomes, from bacterial minicells to endogenous nanovesicles. Exosomes are vesicles released from cells to mediate long distance cell-cell communication [44]. They are emerging delivery vehicles with high potential given their nanosize (50 to 150 nm), biocompatibility, extended blood half-life and low immunogenicity [150] (Figure 3D). The proteins and lipids that exosomes are composed of depends on their cell of origin and can have different therapeutic implications. Exosomes from natural killer (NK) cells contain granzymes, which activate caspases in target cells, and carry a variety of receptors that specifically direct them to ligands expressed on stressed/tumor cells [151,152,153]. These intrinsic cytotoxic, anti-tumor traits have made NK-exosomes (NKExos) ideal therapeutic drug carriers. A recent study demonstrated that NKExos holding Bcl-2 small interfering RNA (siRNA), generated from lentivirally modified NK cells, enhanced apoptosis in estrogen receptor-positive breast cancer cells [154]. The advantages that NKExos provide in terms of tumor specificity, cell uptake and siRNA release make them a promising delivery option for Bcl-2 targeting.

A novel cell-penetrating-peptide equipped exosome has recently been engineered for the optimization of drug delivery via the endocytic pathway. Using Oblimersen as a model, these HepG2 cell derived exosomes, hold a polyarginine peptide (R9) at their surface, EXO-R9-G3139 [155]. R9 is a cell penetrating peptide that serves to enhance exosome cell binding and internalization [155]. HepG2 cells treated with EXO-R9-G3139 for 48 h experienced a 73% decrease in Bcl-2 mRNA expression, and a 70% decrease in Bcl-2 protein expression [155]. Cells treated with EXO-G3139, lacking the R9 peptide, had a 44% and 52% decrease in Bcl-2 mRNA and protein, respectively [155]. Cells treated with G3139 or R9-G3139, on the other hand, showed little difference to untreated cells [155]. This work suggests that exosomes or exosome-like vesicles can be engineered to enhance their drug delivery capacities and may be the next innovative step in oligonucleotide therapies.

## 5. Conclusions

Once identified as a proto-oncogene the targeting of Bcl-2 with ASO therapies was set in motion. Throughout the years much has been learnt about target sequence selection, chemical modifications, stability, delivery and toxicity. Although the progression of several Bcl-2-ASO candidates through clinical trials has ended in failure, due to adverse effects and limited efficacy, the potential of this technology in cancer therapies has not been dispelled. BP1002 remains a promising Bcl-2-ASO in clinical trials. BP1002 builds on the lessons learnt from previous candidates. Its unique P-ethoxy backbone limits the adverse effects of thrombocytopenia and complement activation. Its incorporation into a liposomal delivery vehicle further prolongs its half-life and cellular uptake. Its ability to efficiently suppress Bcl-2 without causing adverse effects nominates it as a valuable contributor in combination therapies and may become a more tolerable alternative to Venetoclax.

The need to safely transport ASO to target cells and allow their intracellular release continues to launch novel delivery technologies. Targeted delivery to cancer cells is a desired goal in the field. Ligand targeted liposomes, which have antibodies or peptides attached to their outer surface, offer site and cell specific delivery. The technology mainly relies on overexpression of specific proteins on tumor cell surfaces [156], however toxicity to normal tissues may still arise as expression of these proteins are found in normal tissues albeit at lower levels. Manufacturing of pharmaceutical grade ligand targeted, ASO-incorporated liposomes might also be complex and expensive. Endogenous and engineered exosomal vesicles also hold great promise [157]. Their biocompatibility and inherent targeted transport makes them exciting ASO carriers with low toxicity and efficient cellular uptake [158]. Nonetheless, large scale purification of exosomal vesicles to meet pharmaceutical quality could be challenging. ASO technology targeting Bcl-2 has come a long way and continues to develop. Although optimizations are still being tested, the potential impact they will have on cancer therapeutics is certainly worth the wait.

## Figures and Tables

**Figure 1 pharmaceutics-14-00097-f001:**
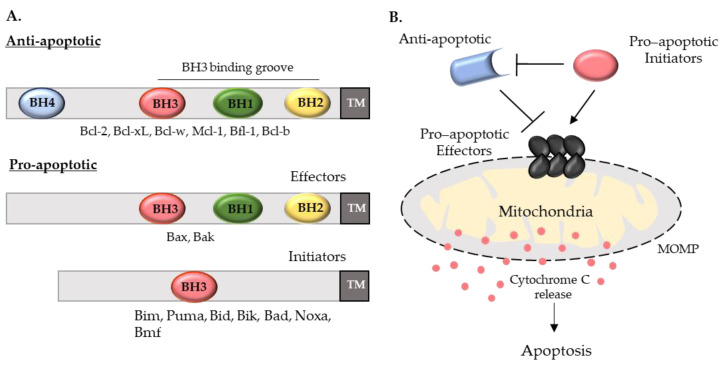
Bcl-2 family proteins and apoptosis overview. (**A**) Bcl-2 family proteins are composed of anti-apoptotic proteins (multiple BH domains) and pro-apoptotic proteins (multi-BH domain effectors and BH3-only initiators). (**B**) Under normal conditions anti-apoptotic proteins inhibit the activity of pro-apoptotic effectors, impeding mitochondrial outer membrane permeabilization (MOMP) and cytochrome C release. Cellular stress activates pro-apoptotic initiators which bind to and inhibit anti-apoptotic proteins, allowing pro-apoptotic effectors to proceed.

**Figure 2 pharmaceutics-14-00097-f002:**
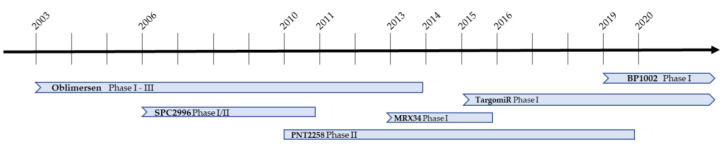
Clinical trial timeline of Bcl-2 ASO and miRNA. Clinical trial start and end dates are marked for each drug. Arrows indicate ongoing clinical trials; blunted arrows represent trials that have been terminated.

**Figure 3 pharmaceutics-14-00097-f003:**
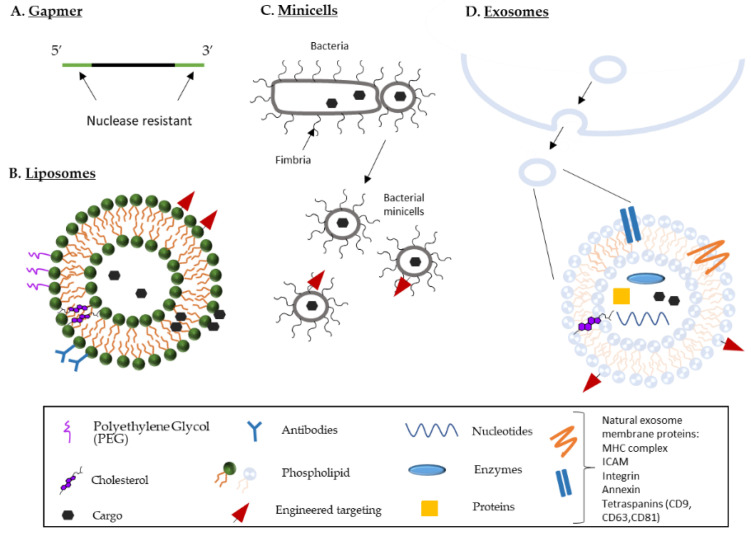
Bcl-2 oligonucleotide delivery. (**A**) Gapmer ASO have nuclease resistant, second, and third generation modifications at the 5′ and 3′ ends. (**B**) Liposomes are composed of phospholipid bilayers (one or multi bilayers; 20–1000 nm in diameter). Depending on the lipids used, liposomes can be cationic, anionic, neutral or pH-sensitive. Hydrophobic cargo is encapsulated within the lipid bilayers, and hydrophilic cargo within the aqueous core or tethered to liposome surface. Additional surface modifications include polyethylene glycol to prolong blood circulation and antibodies or targeting proteins to enhance cancer cell specific delivery. (**C**) Minicells are anucleated, non-dividing, and metabolically active cells (100–300 nm) that bud from abnormally dividing bacteria. Cancer cell targeting can be achieved by the addition of specific peptides or ligands to the minicell surface. (**D**) Exosomes are extracellular vesicles released from cells (50–150 nm). They carry a range of proteins, lipids, and nucleotides, some of which are specific to their cell of origin. Common markers found in exosomes include major histocompatibility complex (MHC), heat shock proteins, tetraspanins, phosphatidylserine, actin, and cholesterol.

## Data Availability

Not applicable.

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
