# Peer review of "Making Sense of Antisense Oligonucleotide Therapeutics Targeting Bcl-2"

_pharmaceutics, 2022, doi:10.3390/pharmaceutics14010097_

Round 1
Reviewer 1 Report
Dysregulation of the B-cell lymphoma 2 (BCL2) expression has been implicated in the pathogenesis of malignancies, primarily tumorigenesis and cellular responses to anti-cancer therapies. Antisense oligonucleotides (ASO) therapeutics as a versatile drug platform represents a promising strategy to target BCL2-driven cancers. In this review article, the authors Gagliardi et al. summarized the historical development of BCL2-targeting ASO and the promising candidates currently in trials.
Overall, the manuscript represents an informative review, which serves the significant interest of readerships in the field. However, the manuscript can benefit from improving the following aspects:
SECTION 1. The narrative in the introduction needs to be more focused and refined. This section should survey sufficient background while maintaining a focus on the rationale of targeting BCL2 in cancers. In addition, as BCL2 is also implicated as a crucial regulator of cellular survival in normal physiology and other non-canonical roles, this section should sufficiently discuss the impact of BCL2 targeting in non-cancerous tissues.
SECTION 2-3. The discussion on BCL2-targeting ASO development needs to be more informative and less descriptive. For example, why was ASO-4625 not able to progress into clinical studies? Also, what was the main reason for Oblimersen to fail to improve overall survival in the trials? These narratives are essential for establishing the manuscript as a more informative scientific review and separating it from descriptive science journalism.
SECTION 4. While antisense oligo technology is the primary scope of this review, it is necessary to compare and contrast ASO with conventional therapy targeting BCL2. Specifically, how does the current ASO compare to the BCL2-targeting small molecule inhibitor such as Venetoclax? The discussion on this topic is limited to describing the challenge of Venetoclax resistance, and it is still unclear whether ASO or ASO adjuvant therapy is expected to be a superior strategy.
Reviewer 2 Report
Well written. Comprehensive summary of the available therapeutics targeting BCL-2.
Reviewer 3 Report
In this manuscript Gagliardi and Ashizawa have reviewed antisense oligonucleotides (ASO) that target Bcl-2 as potential therapeutics for a variety of cancers. The authors are to be commended for a thorough and well presented manuscript.
Major points:
Line 106: the authors state that ‘Bcl-2 was considered undruggable by conventional means’ making it a good candidate for ASO. However, in section 4.1 the authors describe an FDA approved small molecular drug against Bcl-2 (Venetoclax), which appears contradictory. In addition, there is a leap of logic to suggest that the ‘undruggable’ Bcl-2 is therefore a good ASO target. Authors should include more information about the history of Bcl-2-targeting therapeutic development to support their ‘undruggable’ statement. Further, more information is required to explain why intractable targets might be more suited to ASO therapeutics.
Section 2.1
What is the delivery method for the ASO discussed in lines 111-125? This is important information to include as the timeline for intracellular concentrations and cell inhibition levels are compared between different ASOs where the delivery method is explicitly stated.
Section 2.3
What is the in vitro and in vivo data for Oblimersen? This would be helpful to compare data between the discussed ASOs. What is the evidence that Oblimersen is insufficient as a monotherapy?
Section 4.2.1.
What is a liposomal miRNA? (line 349) The delivery methodology is distinct from the drug.
Section 4.2.3
The targeting of delivery is briefly mentioned here (lines 393 and 394) however specificity in delivery of ASOs is a considerable question of the field and the topic could be expanded upon here.
Conclusion:
The submitted conclusion is very brief and is not a thorough evaluation of the literature reviewed. This section should be expanded.
Minor points:
Line 55 – the ‘its’ in the sentence here is ambiguous and requires clarification
Round 2
Reviewer 1 Report
The authors have substantially improved the manuscript and adequately addressed most of my previous concerns.